# Reference-point centering and range-adaptation enhance human reinforcement learning at the cost of irrational preferences

Sophie Bavard [1,2,3], Maël Lebreton[4,5,6], Mehdi Khamassi[7,8], Giorgio Coricelli[9,10] & Stefano Palminteri [1,2,3]

In economics and perceptual decision-making contextual effects are well documented, where decision weights are adjusted as a function of the distribution of stimuli. Yet, in reinforcement learning literature whether and how contextual information pertaining to decision states is integrated in learning algorithms has received comparably little attention. Here, we investigate reinforcement learning behavior and its computational substrates in a task where we orthogonally manipulate outcome valence and magnitude, resulting in systematic variations in state-values. Model comparison indicates that subjects' behavior is best accounted for by an algorithm which includes both reference point-dependence and range-adaptation—two crucial features of state-dependent valuation. In addition, we find that state-dependent outcome valuation progressively emerges, is favored by increasing outcome information and correlated with explicit understanding of the task structure. Finally, our data clearly show that, while being locally adaptive (for instance in negative valence and small magnitude contexts), state-dependent valuation comes at the cost of seemingly irrational choices, when options are extrapolated out from their original contexts.

[1] Laboratoire de Neurosciences Cognitives Computationnelles, Institut National de la Santé et Recherche Médicale, 29 rue d'Ulm, 75005 Paris, France. [2] Département d'Etudes Cognitives, Ecole Normale Supérieure, Paris 75005, France. [3] Institut d'Etudes de la Cognition, Université de Paris Sciences et Lettres, Paris 75005, France. [4] CREED lab, Amsterdam School of Economics, Faculty of Business and Economics, University of Amsterdam, Roetersstraat 11, Amsterdam 1018 WB, The Netherlands. [5] Amsterdam Brain and Cognition, University of Amsterdam, Amsterdam 1018 WB, The Netherlands. [6] Swiss Centre for Affective Sciences, University of Geneva, 24 rue du Général-Dufour, Geneva 1205, Switzerland. [7] Institut des Systèmes Intelligents et Robotiques, Centre National de la Recherche Scientifique, 4 place Jussieu, 75005 Paris, France. [8] Institut des Sciences de l'Information et de leurs Interactions, Sorbonne Universités, 3 rue Michel-Ange, Paris 75794, France. [9] Department of Economics, University of Southern California, Los Angeles, CA 90007, USA. [10] Centro Mente e Cervello, Università di Trento, corso Bettini 21, Rovereto 38068, Italy. These authors contributed equally: Sophie Bavard, Maël Lebreton. Correspondence and requests for materials should be addressed to S.P. (email: stefano.palminteri@ens.fr)

In everyday life, our decision-making abilities are solicited in situations that range from the most mundane (choosing how to dress, what to eat, or which road to take to avoid traffic jams) to the most consequential (deciding to get engaged, or to give up on a long-lasting costly project). In other words, our actions and decisions result in outcomes, which can dramatically differ in terms of affective valence (positive vs. negative) and intensity (small vs. big magnitude). These two features of the outcome value are captured by different psychological concepts—affect vs. salience—and by different behavioral and physiological manifestations (approach/avoidance vs. arousal/energization levels)[1–3].

In ecological environments, where new options and actions are episodically made available to a decision-maker, both the valence and magnitude associated with the newly available option and action outcomes have to be learnt from experience. The reinforcement-learning (RL) theory offers simple computational solutions, where the expected value (product of valence and magnitude) is learnt by trial-and-error, thanks to an updating mechanism based on prediction error correction[4,5]. RL algorithms have been extensively used during the past couple of decades in the field of cognitive neuroscience, because they parsimoniously account for behavioral results, neuronal activities in both human and non-human primates, and psychiatric symptoms induced by neuromodulatory dysfunction[6–10].

However, this simple RL model is unsuited to be used as is in ecological contexts[11,12]. Rather, similarly to the perceptual and economic decision-making domains, growing evidence suggests that reinforcement learning behavior is sensitive to contextual effects[13–16]. This is particularly striking in loss-avoidance contexts, where an avoided-loss (objectively an affectively neural event) can become a relative reward if the decision-maker has frequently experienced losses in the considered environment. In that case, the decision-maker's knowledge about the reward distribution in the recent history or at a specific location, affects her perception of the valence of outcomes. Reference-dependence, i.e., the evaluation of outcomes as gains or losses relative to a temporal or spatial reference point (context), is one of the fundamental principles of prospect theory and behavioral economics[17]. Yet, only recently have theoretical and experimental studies in animal and human investigated this reference-dependence in RL[18–20]. These studies have notably revealed that reference-dependence can significantly improve learning performances in contexts of negative valence (loss-avoidance), but at the cost of generating post-learning inconsistent preferences[18,19].

In addition to this valence reference-dependence, another important contextual effect that may be incorporated in ecological RL algorithms is range adaptation. At the behavioral level, it has long been known that our sensitivity to sensory stimuli or monetary amounts is not the same across different ranges of intensity/magnitude[21,22]. These findings have recently paralleled with the description of neuronal range adaptation: in short, the need to provide efficient coding of information in various ranges of situations entails that the firing rate of neuron adapts to the distributional properties of the variable being encoded[23]. Converging pieces of evidence have recently confirmed neuronal range-adaptation in economic and perceptual decision-making, although its exact implementation remains debated[24–27].

Comparatively, the existence of behavioral and neural features of range-adaptation has been less explored in RL, where it could critically affect the coding of outcome magnitude. In the RL framework the notion of context, which is more prevalent in the economic or perception literatures, is embodied in the notion of state. In the RL framework the environment is defined as a collection of discrete states, where stimuli are encountered, decisions are made and outcomes are collected. Behavioral and neural manifestations of context-dependence could therefore be achieved by (or reframed as) state-dependent processes.

Here, we hypothesized that in human RL, the trial-by-trial learning of option and action values is concurrently affected by reference-point centering and range adaptation. To test this hypothesis and investigate the computational basis of such state-dependent learning, we adapted a well-validated RL paradigm[19,28], to include orthogonal manipulations of outcome valence and outcome magnitude.

Over two experiments we found that human RL behavior is consistent with value-normalization, both in terms of state-based reference-dependence and range-adaptation. To better characterize this normalization process at the algorithmic level, we compared several RL algorithms, which differed in the extent and in the way they implement state-dependent valuation (reference-dependence and range adaptation). In particular, we contrasted models implementing full, partial or no value normalization[29]. We also evaluated models implementing state-dependent valuation at the decision stage (as opposed to the outcome evaluation stage) and implementing marginally decreasing utility (as proposed by Bernoulli)[22]. Overall, the normalization process was found to be partial, to occur at the valuation level, to progressively arise during learning and to be correlated with explicit understanding of the task structure (environmental). Finally, while being optimal in an efficient coding perspective, this normalization leads to irrational preference when options are extrapolated out from their original learning context.

## Results

**Behavioral paradigm to challenge context-dependence.** Healthy subjects performed two variants of a probabilistic instrumental learning task with monetary rewards and losses. In those two variants, participants saw at each trial a couple of abstract stimuli (options), which were probabilistically paired with good or bad outcomes, and had to select the one they believed would be most beneficial for their payoff. The options were always presented in fixed pairs, which defined stable choice contexts. These contexts were systematically manipulated, so as to implement a $2 \times 2$ factorial design across two qualities of the option outcomes: outcome valence (reward or loss) and outcome magnitude (big: 1€; or small: 10c). In all contexts, the two options were associated with different, stationary, outcome probabilities (75% or 25%). The 'favorable' and 'unfavorable' options differ in their net expected value. The favorable option in the reward and big magnitude context is paired with a reward of 1€ with probability 75%, while the unfavorable option only 25% of the time. Likewise, the favorable option in the loss and small magnitude context is paired with a loss of 10 cents with probability 25%, while the unfavorable option 75% of the time (Fig. 1). Subjects therefore had to learn to choose the options associated either with highest reward probability or those associated with lowest loss probability. After the last learning session, subjects performed a transfer test in which they were asked to indicate the option with the highest value, in choices involving all possible binary combinations—that is, including pairs of options that had never been associated during the task. Transfer test choices were not followed by feedback, to not interfere with subjects' final estimates of option values. In the second variant of the experiment, an additional factor was added to the design: the feedback information about the outcomes (partial or complete) was manipulated to make this variant a $2 \times 2 \times 2$ factorial design. In the partial context, participants were only provided with feedback about the option they chose, while in the complete context, feedback about the outcome of the non-chosen option was also provided.

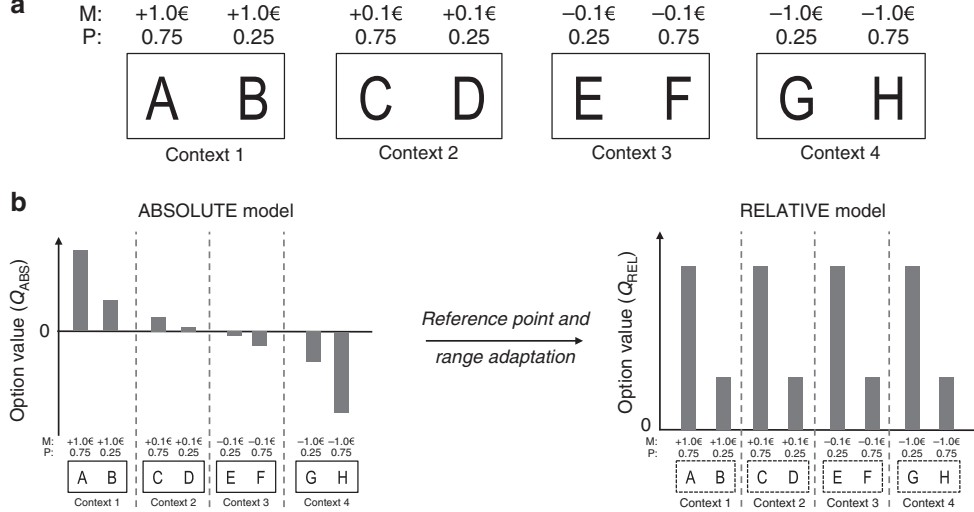

**Fig. 1** Experimental design and normalization process. **a** Learning task with four different contexts: reward/big, reward/small, loss/small, and loss/big. Each symbol is associated with a probability (*P*) of gaining or losing an amount of money or magnitude (*M*). *M* varies as a function of the choice contexts (reward seeking: +1.0€ or +0.1€; loss avoidance: −1.0€ or −0.1€; small magnitude: +0.1€ or −0.1€; big magnitude: +1.0€ or −1.0€). **b** The graph schematizes the transition from absolute value encoding (where values are negative in the loss avoidance contexts and smaller in the small magnitude contexts) to relative value encoding (complete adaptation as in the RELATIVE model), where favorable and unfavorable options have similar values in all contexts, thanks to both reference-point and range adaptation

**Table 1 Correct choice rate of the learning sessions as a function of task factors in Experiments 1, 2 and both experiments**

|  | Experiment 1 (N = 20) | | Experiment 2 (N = 40) | | Both experiments (N = 60) | |
|---|---|---|---|---|---|---|
|  | *F*-val | *P*-val | *F*-val | *P*-val | *F*-val | *P*-val |
| Val | 0.002 | 0.969 | 0.285 | 0.597 | 0.167 | 0.684 |
| Inf | – | – | 7.443 | 0.0095** | – | – |
| Mag | 4.872 | 0.0398* | 4.267 | 0.0456* | 9.091 | 0.00378** |
| Val × Inf | – | – | 1.037 | 0.315 | – | – |
| Val × Mag | 4.011 | 0.0597 | 0.08 | 0.779 | 1.755 | 0.19 |
| Inf × Mag | – | – | 0.006 | 0.939 | — | — |
| Val × Inf × Mag | – | – | 0.347 | 0.559 | — | — |

**P < 0.01; *P < 0.05, *t*-test

**Outcome magnitude moderately affects learning performance**. In order to characterize the learning behavior of participants in our tasks, we first simply analyzed the correct response rate in the learning sessions, i.e., choices directed toward the most favorable stimulus (i.e., associated with the highest expected reward or the lowest expected loss). In all contexts, this average correct response rate was higher than chance level 0.5, signaling significant instrumental learning effects ($T(59) = 16.6$, $P < 0.001$). We also investigated the effects of our main experimental manipulations (outcome valence (reward/loss), outcome magnitude (big/small), and feedback information (partial/complete, Experiment 2 only)) (Table 1). Because there was no significant effect of the experiment (i.e., when explicitly entered as factor 'Experiment': $F(59) = 0.96$, $P > 0.3$), we pooled the two experiments to assess the effects of common factors (outcome valence and magnitude). Replicating previous findings[19], we found that the outcome valence did not affect learning performance ($F(59) = 0.167$, $P > 0.6$), and that feedback information significantly modulated learning in Experiment 2 ($F(39) = 7.4$, $P < 0.01$). Finally, we found that the outcome magnitude manipulation,

which is a novelty of the present experiments, had a significant effect on learning performance ($F(59) = 9.09$, $P < 0.004$); Post-hoc test confirmed that across both experiments subjects showed significantly higher correct choice rate in the big-magnitude compared with the small-magnitude contexts ($T(59) > 3.0$, $P < 0.004$), and similar correct choice rate in the reward compared to the losses contexts ($T(59) = 0.41$, $P > 0.13$).

**Transfer test choices do not follow expected values**. Following the analytical strategy used in previous studies[18,19], we next turned to the results from the transfer test, and analyzed the pattern of correct choice rates, i.e., the proportion of choices directed toward the most favorable stimulus (i.e., associated with the highest expected reward or the lowest expected loss). Overall, the correct choice rate in the transfer was significantly higher than chance, thus providing evidence of significant value transfer and retrieval ($T(59) > 3.0$, $P < 0.004$). We also analyzed how our experimental factors (outcome valence (reward/loss), outcome magnitude (big/small) and option favorableness (i.e., being the symbol the most favorable of its pair during the learning sessions) influenced the choice rate per symbol. The choice rate per symbol is the average frequency with which a given symbol is chosen in the transfer test, and can therefore be taken as a measure of the subjective preference for a given option. Consistent with significant value transfer and retrieval, the ANOVA revealed significant effects of outcome valence ($F(59) = 76$, $P < 0.001$) and option correctness ($F(59) = 203.5$, $P < 0.001$) indicating that—in average—symbols associated with favorable outcomes were preferred compared to symbols associated with less favorable ones (Table 2). However, and in line with what we found in simpler contexts[19,28], the analysis of the transfer test revealed that option preference did not linearly follow the objective ranking based on their absolute expected value (probability(outcome) × magnitude (outcome)). For example, the favorable option of the reward/small context was chosen more often than the less favorable option of the reward/big context ($0.71 \pm 0.03$ vs. $0.41 \pm 0.04$; $T(59) = 6.43$, $P < 0.0001$). Similarly, the favorable option of the loss/small magnitude context was chosen more often than the less favorable option of the reward/small context ($0.42 \pm 0.03$ vs. 0.56

**Table 2 Symbol choice rate of the transfer test as a function of task factors and option correctness in Experiments 1, 2 and both experiments**

| | Experiment 1 ($N = 20$) | | Experiment 2 ($N = 40$) | | Both experiments ($N = 60$) | |
| --- | --- | --- | --- | --- | --- | --- |
| | **F-val** | **P-val** | **F-val** | **P-val** | **F-val** | **P-val** |
| Valence | 33.42 | 1.43e−05*** | 43.78 | 7.23e−08*** | 76 | 3.38e−12*** |
| Favorableness | 57.66 | 3.6e−07*** | 149.5 | 6.46e−15*** | 203.5 | <2e−16*** |
| Magnitude | 2.929 | 0.103 | 4.225 | 0.0466* | 0.525 | 0.472 |
| Val × Fav | 4.039 | 0.0589 | 6.584 | 0.0142* | 10.8 | 0.00171** |
| Val × Mag | 11.68 | 0.00289** | 3.565 | 0.0665 | 11.55 | 0.00122** |
| Fav × Mag | 10.8 | 0.00388** | 0.441 | 0.51 | 4.131 | 0.0466* |
| Val × Fav × Mag | 8.241 | 0.00979** | 1.529 | 0.224 | 7.159 | 0.00964** |

***P < 0.001; *P < 0.05; **P < 0.01; t-test

± 0.03; $T(59) = 2.88$, $P < 0.006$). Crucially, while the latter value inversion reflects reference-point dependence, as shown in previous studies[19,28], the former effect is new and could be a signature of a more global range-adaptation process. To verify that these value inversions were not only observed at the aggregate level (i.e., were not an averaging artifact), we analyzed the transfer test choice rate for each possible comparison. Crucially, analysis of the pairwise choices confirm value inversion also for direct comparisons.

**Delineating the computational hypothesis.** Although these overall choice patterns appear puzzling at first sight—since they would be classified as "irrational" from the point of view of the classical economic theory based on absolute values[30]—we previously reported that similar seemingly irrational behavior and inconsistent results could be coherently generated and explained by state-dependent RL models. To hypothesize this reasoning, we next turned to computational modeling to provide a parsimonious explanation of the present results.

To do so, we fitted the behavioral data with several variations of standard RL models (see Methods). The first model is a standard Q-learning algorithm, referred to as ABSOLUTE. The second model is a modified version of the Q-learning model that encodes outcomes in a state-dependent manner:

$$R_{\mathrm{REL},t} = \frac{R_{\mathrm{ABS},t}}{|V_t(s)|} + \max\left\{0, \frac{-V_t(s)}{|V_t(s)|}\right\} \quad (1)$$

where the state value $V(s)$ is initialized to 0, takes the value of the first non-zero (chosen or unchosen) outcome in each context $s$, and then remains stable over subsequent trials. The first term of the question implements range adaptation (divisive normalization) and the second term reference point-dependence (subtractive normalization). As a result, favorable/unfavorable outcomes are encoded in a binary scale, despite their absolute scale. We refer to this model as RELATIVE, while highlighting here that this model extends and generalizes the so-called "RELATIVE model" employed in a previous study, since the latter only incorporated a reference-point-dependence subtractive normalization term, and not a range adaptation divisive normalization term[19].

The third model, referred to as HYBRID, encodes the reward as a weighted sum of an ABSOLUTE and a RELATIVE reward:

$$R_{\mathrm{HYB},t} = \omega * R_{\mathrm{REL},t} + (1 - \omega) * R_{\mathrm{ABS},t} \quad (2)$$

The weight parameter ($\omega$) of the HYBRID model quantifies at the individual level the balance between absolute ($\omega = 0.0$) and relative value encoding ($\omega = 1.0$).

The fourth model, referred to as the UTILITY model, implements the economic notion of marginally decreasing subjective utility[17,22]. Since our task included only two non-zero outcomes, we implemented the UTILITY model by scaling the big magnitude outcomes ($|1€|$) with a multiplicative factor ($0.1 < v < 1.0$).

Finally, the fifth model, referred to as the POLICY model, normalizes (range adaptation and reference point correction) values at the decision step (i.e., in the softmax), where the probability of choosing 'a' over 'b' is defined by

$$P_t(s, a) = \frac{1}{1 + e^{\left(\frac{Q_t(s,b) - Q_t(s,a)}{Q_t(s,b) + Q_t(s,a)} * \frac{1}{\beta}\right)}} \quad (3)$$

**Model comparison favors the HYBRID model.** For each model, we estimated the optimal free parameters by likelihood maximization. The Bayesian Information Criterion (BIC) was then used to compare the goodness-of-fit and parsimony of the different models. We ran three different optimization and comparison procedures, for the different phases of the experiments: learning sessions only, transfer test only, and both tests. Thus we obtained a specific fit for each parameter and each model in the learning sessions, transfer test, and both.

Overall (i.e., across both experiments and experimental phases), we found that the HYBRID model significantly better accounted for the data compared to the RELATIVE, the ABSOLUTE, the POLICY, and the UTILITY models (HYB vs. ABS $T(59) = 6.35$, $P < 0.0001$; HYB vs. REL $T(59) = 6.07$, $P < 0.0001$; HYB vs. POL $T(59) = 6.79$, $P < 0.0001$; HYB vs. UTY $T(59) = 2.72$, $P < 0.01$). This result was robust across experiments and across experimental sessions (learning sessions vs. transfer test) (Table 3). In the main text we focus on discussing the ABSOLUTE and the RELATIVE models, which are nested within the HYBRID and therefore represent extreme cases (absent or complete) of value normalization. We refer to the Supplementary Methods for a detailed analysis of the properties of the POLICY and the UTILITY models (Supplementary Figure 1), and additional model comparison (Supplementary Table 1).

**Model simulations falsify the ABSOLUTE and RELATIVE models.** Although model comparison unambiguously favored the HYBRID model, we next aimed to falsify the alternative models, using simulations[31]. To do so, we compared the correct choice rate in the learning sessions to the model predictions of the three main models (ABSOLUTE, RELATIVE, and HYBRID). We generated for each model and for each trial $t$ the probability of choosing the most favorable option, given the subjects' history of choices and outcomes, using the individual best-fitting sets of

**Table 3 BICs as a function of the dataset used for parameter optimization (Learning sessions, Transfer test or Both) and the computational model**

| | Experiment 1 (N = 20) | | | Experiment 2 (N = 40) | | | Both experiments (N = 60) | | |
|---|---|---|---|---|---|---|---|---|---|
| | Learning sessions (nt = 160) | Transfer test (nt = 112) | Both (nt = 272) | Learning sessions (nt = 160) | Transfer test (nt = 112) | Both (nt = 272) | Learning sessions (nt = 160) | Transfer test (nt = 112) | Both (nt = 272) |
| ABSOLUTE (df = 2/3) | 179.8 ± 5.9 | 113.6 ± 5.7 | 295.1 ± 9.9 | 190.9 ± 5.9 | 126.9 ± 4.1 | 325.4 ± 6.5 | 187.2 ± 3.8 | 122.4 ± 3.4 | 315.3 ± 5.6 |
| RELATIVE (df = 2/3) | 193.3 ± 4.5 | 135.8 ± 5.1 | 329.6 ± 8.0 | 185.1 ± 5.6 | 121.1 ± 4.0 | 306.0 ± 7.3 | 187.9 ± 4.0 | 126.0 ± 3.3 | 313.9 ± 5.7 |
| HYBRID (df = 3/4) | 178.3 ± 6.0 | 109.3 ± 5.0 | 284.6 ± 9.1 | 181.5 ± 5.8 | 105.8 ± 4.1 | 290.5 ± 8.0 | 180.5 ± 4.3 | 106.9 ± 3.2 | 288.5 ± 6.1 |
| POLICY (df = 2/3) | 185.4 ± 6.9 | 123.7 ± 6.3 | 311.0 ± 12.2 | 190.1 ± 4.9 | 139.4 ± 3.9 | 334.6 ± 6.5 | 188.5 ± 3.9 | 134.2 ± 3.4 | 326.7 ± 6.0 |
| UTILITY (df = 3/4) | 173.9 ± 6.5 | 107.5 ± 6.3 | 282.2 ± 10.8 | 183.4 ± 5.6 | 123.1 ± 4.5 | 310.1 ± 7.1 | 180.2 ± 4.3 | 117.9 ± 3.8 | 300.8 ± 6.2 |

Nt, number of trials; df, degree of freedom

parameters. Concerning the learning sessions, we particularly focused on the magnitude effect (i.e., the difference in performance between big and small magnitude contexts). As expected, the ABSOLUTE model exacerbates the observed magnitude effect (simulations vs. data, $T(59) = 5.8$, $P < 0.001$). On the other side, the RELATIVE model underestimates the actual effect (simulations vs. data, $T(59) = 3.0$, $P < 0.004$). Finally (and unsurprisingly), the HYBRID model manages to accurately account for the observed magnitude effect ($T(59) = 0.93$, $P > 0.35$) (Fig. 2a, b). We subsequently compared the choice rate in the transfer test to the three models' predictions. Both the ABSOLUTE and the RELATIVE models failed to correctly predict choice preference in the transfer test (Fig. 2c). Crucially, both models failed to predict the choice rate of intermediate value options. The ABSOLUTE model predicted a quite linear option preference, predicting that the transfer test choice rate should be highly determined by the expected utility of the options. On the other side, the RELATIVE model's predictions of the transfer test option preferences were uniquely driven by the option context-dependent favorableness. Finally, choices predicted by the HYBRID model accurately captured the observed option preferences by predicting both an overall correlation between preferences and expected utility and the violation of the monotony of this relation concerning intermediate value options (Figs. 2d, 3). To summarize, and similarly to what was observed in previous studies[18,19,29], choices in both the learning and transfer test could not be explained by assuming that option values are encoded in an absolute manner, nor by assuming that they are encoded in a fully context-dependent manner, but are consistent with a partial context dependence. In the subsequent sections we analyze the factors that affect value contextualization both within and between subjects.

**Relative value encoding emerges during learning**. Overall we found that a weighted mixture of absolute and relative value encoding (the HYBRID model) better explained the data compared to the "extreme" ABSOLUTE or RELATIVE models. However, this model comparison integrates over all the trials, leaving open the possibility that, while on average subjects displayed no neat preference for either of the two extreme models, this result may arise from averaging over different phases in which one of the models could still be preferred. To test this hypothesis, we analyzed the trial-by-trial likelihood difference between the RELATIVE and the ABSOLUTE model. This quantity basically measures which model better predicts the data in a given trial: if positive, the RELATIVE model better explains the data, if negative, the ABSOLUTE model does. We submitted the trial-by-trial likelihood difference during a learning session to

a repeated measure ANOVA with 'trial' (1:80) as within-subject factor. This analysis showed a significant effect of trial indicating that the evidence for the RELATIVE and the ABSOLUTE model evolves over time ($F(79) = 6.2$, $P < 2e-16$). Post-hoc tests revealed two big clusters of trials with non-zero likelihood difference: a very early cluster (10 trials from the 4th to the 14th) and a very late one (17 trials from the 62nd to the 78th). To confirm this results, we averaged across likelihood difference in the first half (1:40 trials) and in the second half (41:80 trials). In the first half we found this differential to be significantly negative, indicating that the ABSOLUTE model better predicted subjects' behavior ($T(59) = 2.1$, $P = 0.036$). In contrast, in the second half we found this differential to be significantly positive, indicating that the RELATIVE model better predicted subjects' behavior ($T(59) = 2.1$, $P = 0.039$). Furthermore, a direct comparison between the two phases also revealed a significant difference ($T(59) = 3.9$, $P = 0.00005$) (Fig. 4a, b). Finally, consistent with a progressively increasing likelihood of the RELATIVE compared the ABSOLUTE model during the learning sessions, we found that the weight parameter ($\omega$) of the HYBRID model obtained from the transfer test ($0.50 \pm 0.05$) was numerically higher compared to that of the learning sessions ($0.44 \pm 0.05$) (Table 4).

**Counterfactual information favors relative value learning**. The two experiments differed in that in the second one (Experiment 2) half of the trials were complete feedback trials. In complete feedback trials, subjects were presented with the outcomes of both the chosen and the forgone options. In line with the observation that information concerning the forgone outcome promotes state-dependent valuation both at the behavioral and neural levels[18,32], we tested whether or not the presence of such "counterfactual" feedbacks affects the balance between absolute and relative value learning. To do so, we compared the negative log-likelihood difference between the RELATIVE and the ABSOLUTE model separately for the two experiments. Note that since the two models have the same number of free parameters, they can be directly compared using the log-likelihood. In Experiment 2 (where 50% of the trials were "complete feedback" trials) we found this differential to be significantly positive, indicating that the RELATIVE model better fits the data ($T(39) = 2.5$, $P = 0.015$). In contrast, in Experiment 1 (where 0% of the trials were "complete feedback" trials), we found this differential to be significantly negative, indicating that the ABSOLUTE model better fits the data ($T(19) = 2.9$, $P = 0.001$). Furthermore, a direct comparison between the two experiments also revealed a significant difference ($T(58) = 3.9$, $P = 0.0002$) (Fig. 4c). Accordingly, we also found the weight parameter ($\omega$) of

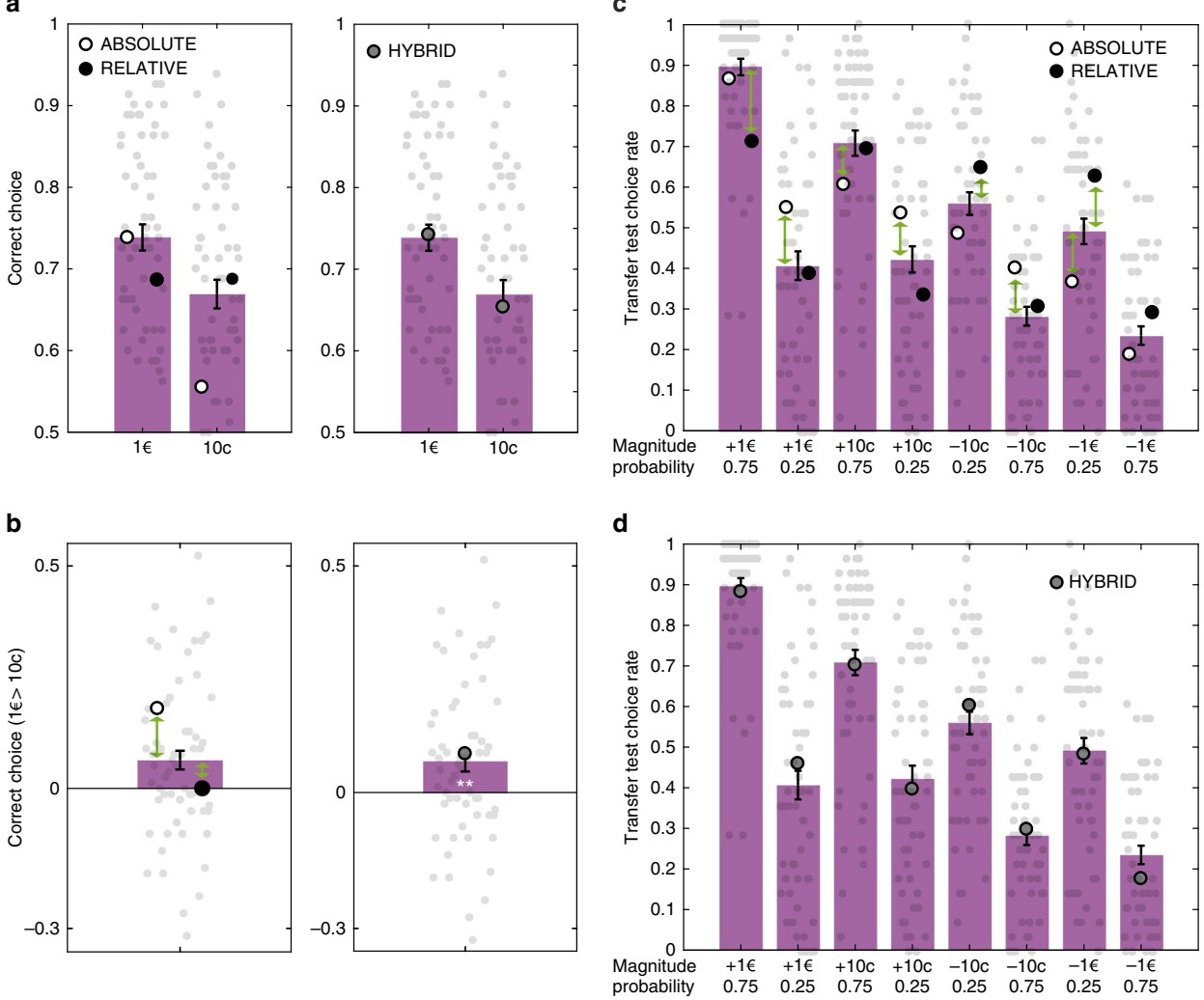

**Fig. 2** Behavioral results and model simulations. **a** Correct choice rate during the learning sessions. **b** Big magnitude contexts' minus small magnitude contexts' correct choice rate during the learning sessions. **c** and **d** Choice rate in the transfer test. Colored bars represent the actual data. Big black (RELATIVE), white (ABSOLUTE), and gray (HYBRID) dots represent the model-predicted choice rate. Small light gray dots above and below the bars represent individual subjects (N = 60). White stars indicate significant difference compared to zero. Error bars represent s.e.m. **P < 0.01, t-test. Green arrows indicate significant differences between actual and predicted choices at P < 0.001, t-test

the HYBRID model to be significantly higher in Experiment 2 compared to Experiment 1 ($T(58) = 2.8$, $P = 0.007$) (Fig. 4d). Finally, consistently with reduced relative value learning, we found that the correct choice difference between the 1€ and the 0.1€ contexts in Experiment 1 (mean: +0.10; range: −0.24/+0.51) was 189.5% of that observed in Experiment 2 (mean: +0.05; range: −0.32/+0.40).

**Explicit grasp of task structure links to relative valuation.** In our learning protocol the fact that options were presented in fixed pairs (i.e., contexts) has to be discovered by subjects, because the information was not explicitly given in the instructions and the contexts were not visually cued. In between the learning and the transfer phases subjects were asked whether or not they believed that options were presented in fixed pairs and how many pairs there were (in the second session). Concerning the first question ("fixed pairs"), 71.7% of subjects responded correctly. Concerning the second question ("pairs number"), 50.0% of subjects responded correctly and the average number of pairs was $3.60 ± 0.13$, which significantly underestimated the true value (four: $T(59) = 3.0$, $P = 0.0035$). To test whether or not the explicit

knowledge of the subdivision of the learning task in discrete choice contexts was correlated with the propensity to learn relative values, we calculated the correlation between the number of correct responses in the debriefing (0, 1, or 2) and the weight parameter ($\omega$) of the HYBRID model. We found a positive and significant correlation ($R^2 = 0.11$, $P = 0.009$) (direct comparison of the weight parameter ($\omega$) between subjects with 0 vs. 2 correct responses in the debriefing: $T(37) = 2.8$, $P = 0.0087$) (Fig. 4e). To confirm this result, we ran the reciprocal analysis, by splitting subjects into two groups according to their weight parameter and we found that subjects with $\omega > 0.5$ had a significantly higher number of correct responses in the debriefing compared to subjects with $\omega < 0.5$ ($T(58) = 3.0$, $P = 0.0035$) (Fig. 4f).

**Rational and irrational consequences of relative valuation.** Previous behavioral analyses, as well as model comparison results, showed that a mixture of relative and absolute value learning (the HYBRID model) explained subjects' behavior. In particular, during the learning sessions, subjects displayed a correct choice difference between the 1€ and the 0.1€ contexts smaller than that predicted by the ABSOLUTE model. During the transfer test, the

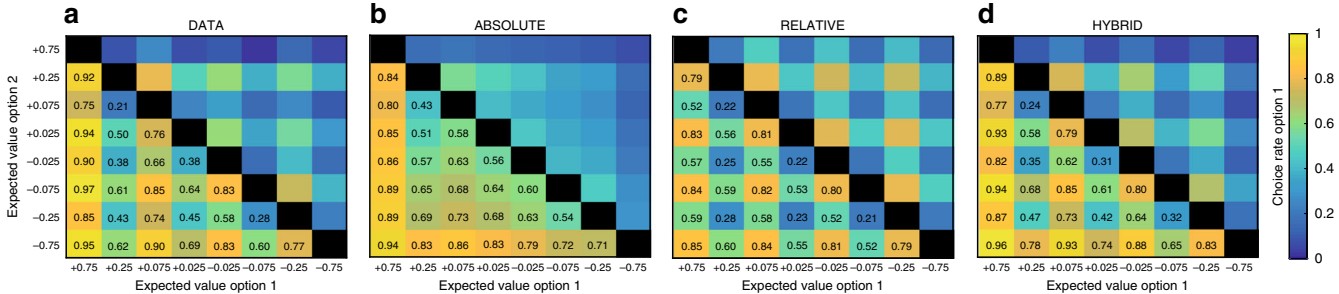

**Fig. 3** Transfer test behavioral results and model simulations. Colored map of pairwise choice rates during the transfer test for each symbol when compared to each of the seven other symbols, noted here generically as 'option 1' and 'option 2'. Comparisons between the same symbols are undefined (black squares). **a** Experimental data, **b** ABSOLUTE model, **c** RELATIVE model, and **d** HYBRID model

**Fig. 4** Computational properties and behavioral correlates of value normalization. **a** Likelihood difference (from model fitting) between the RELATIVE and the ABSOLUTE models over the 80 trials of the task sessions for both experiments ($N = 60$). A negative likelihood difference means that the ABSOLUTE model is the best-fitting model for the trial and a positive likelihood difference means that the RELATIVE model is the best-fitting model for the trial. Green dots: likelihood difference significantly different from 0 ($P < 0.05$, $t$-test). **b** Likelihood difference between the RELATIVE and the ABSOLUTE models over the first part of the task (40 first trials) and the last part (40 last trials) for both experiments. **c** Likelihood difference between the RELATIVE and the ABSOLUTE models for the two experiments. A negative likelihood difference means that the ABSOLUTE model is the best-fitting model for the experiment and a positive likelihood difference means that the RELATIVE model is the best-fitting model for the experiment. **d** Subject-specific free parameter weight ($\omega$) comparison for the two experiments. **e** Subject-specific free parameter weight ($\omega$) as a function of correct debriefing for the two questions ("fixed pairs" and "number of pairs"). **f** Debriefing as a function of the weight parameter. Small light gray dots above and below the bars in **a**–**f** represent individual subjects ($N = 60$). **g** and **h** Correct choice rate as a function of subjects' weight parameter in the learning sessions and the transfer test for both Experiments 1 and 2. One dot corresponds to one participant ($N = 60$); green lines represent the linear regression calculations. Error bars represent s.e.m. ***$P < 0.001$, **$P < 0.01$, *$P < 0.05$, $t$-test

**Table 4 Model parameters of the HYBRID model as a function of the dataset used for parameter optimization (learning sessions, transfer test or Both) and the computational model**

| | Experiment 1 (N = 20) | | | Experiment 2 (N = 40) | | | Both experiments (N = 60) | | |
|---|---|---|---|---|---|---|---|---|---|
| | Learning sessions | Transfer test | Both | Learning sessions | Transfer test | Both | Learning sessions | Transfer test | Both |
| $\beta$ | 0.15 ± 0.04 | 0.12 ± 0.03 | 0.09 ± 0.02 | 0.30 ± 0.11 | 0.13 ± 0.04 | 0.17 ± 0.04 | 0.25 ± 0.08 | 0.13 ± 0.03 | 0.15 ± 0.03 |
| $\alpha_F$ | 0.25 ± 0.06 | 0.30 ± 0.08 | 0.14 ± 0.04 | 0.23 ± 0.04 | 0.34 ± 0.07 | 0.20 ± 0.04 | 0.24 ± 0.04 | 0.33 ± 0.05 | 0.18 ± 0.03 |
| $\alpha_C$ | — | — | — | 0.16 ± 0.04 | 0.25 ± 0.05 | 0.16 ± 0.03 | — | — | — |
| $\omega$ | 0.29 ± 0.07 | 0.34 ± 0.06 | 0.34 ± 0.06 | 0.52 ± 0.06 | 0.58 ± 0.06 | 0.58 ± 0.05 | 0.44 ± 0.05 | 0.50 ± 0.05 | 0.50 ± 0.04 |

response pattern indicated, consistent with the RELATIVE model, "correct" options with lower expected utility were often preferred to "incorrect" options with higher expected utility. To formally test the hypothesis that relative value learning is positively associated with correct choice in the learning phase (i.e., rational) and negatively associated with correct choice (i.e., choice of the option with the highest absolute value) in the transfer phase (i.e., irrational), we tested the correlation between correct choice rates in these two phases and the weight parameter ($\omega$), which quantifies the balance between the ABSOLUTE ($\omega = 0.0$) and RELATIVE models ($\omega = 1.0$). Consistent with this idea we found a positive and significant correlation between the weight parameter and the correct choice rate in the 0.1€ contexts ($R^2 = 0.19$, $P = 0.0005$) and a negative and significant correlation between the same parameter and the correct choice rate in the transfer test ($R^2 = 0.42$, $P = 0.00000003$) (Fig. 4g, h). This means that, the better a subject was at picking the correct option during the learning phase (rational behavior), the least often she would pick the option with the highest absolute value during the test phase (irrational behavior).

## Discussion

In the present paper, we investigated state-dependent valuation in human reinforcement learning. In particular, we adapted a task designed to address the reference-dependence[19] to include an additional manipulation of the magnitude of outcomes, in order to investigate range-adaptation[26]. In the learning sessions, analyses of behavioral data showed that the manipulation of outcome magnitude had a significant effect on learning performance, with high-magnitude outcomes inducing better learning compared to low-magnitude outcomes. On the contrary, and in line with what we reported previously[19], the manipulation of outcome valence had no such effect. In the transfer test, participants exhibited seemingly irrational preferences, sometimes preferring options that had objectively lower expected values than other options. Crucially, these irrational preferences are compatible with state-dependent valuation.

State-dependent (or context-dependent) valuation has been ascribed to a large number of different behavioral, neural and computational manifestations[16]. Under this rather general umbrella, reference-dependence and range-adaptation constitute two specific, and in principle dissociable, mechanisms: on the one hand, reference-dependence is the mechanism through which, in a context where monetary losses are frequent, loss avoidance (an affective neural event) is experienced as a positive outcome. On the other hand, range-adaptation is the mechanism through which, in contexts with different outcome magnitudes (i.e.,

different affective saliency), high-magnitude and low-magnitude outcomes are experienced similarly.

In order to formally and quantitatively test for the presence of these two components of state-dependent valuation in our experimental data, we used computational modeling. Our model space included two 'extreme' models: the ABSOLUTE and the RELATIVE models. The ABSOLUTE model learns the context-independent—absolute—value of available options. In contrast, the RELATIVE model implements both reference-dependence and range-adaptation ('full' adaptation[29]). These two 'extreme' models predict radically different choice patterns in both the learning sessions and the transfer test. While the ABSOLUTE model predicts a big effect of outcome magnitude in the learning sessions and rational preferences in the transfer test, the RELATIVE model predicts no magnitude effect and highly irrational preferences in the transfer test. Specifically, according to the RELATIVE model, the choices in the transfer test are not affected by the outcome valence or by the outcome magnitude, but dominated by options' context-dependent favorableness factor. Comparison between model simulations and experimental data falsified both models[31], since in both the learning sessions and in the transfer test, subjects performance lied in between the predictions of the ABSOLUTE and RELATIVE models. To account for this pattern we designed a HYBRID model. The HYBRID model implements a trade-off between the absolute and relative learning modules, which is governed by an additional free parameter ('partial adaptation'[29]). Owing to this partial adaptation, the HYBRID model accurately accounts for the performance in the learning sessions and for the preferences expressed in the transfer test, including the preference inversion patterns.

Using model comparison, we attempted to provide a specific description of the process at stake in our task, and ruled out alternative accounts of normalization. Crucially, normalization can be implemented as an adaptation over time of the valuation mechanism to account for the distribution of option values encountered in successive choices, or as a time-independent decision mechanism limited to the values of options considered in one choice event[24,33]. In the present case, model comparison favored the HYBRID model, which implements a time-adapting value normalization against the POLICY model, which implements a time-independent decision normalization. This result derives from the fact that during the learning sessions, the POLICY model uses a divisive normalization at the moment of choice to level the learning performance in different contexts (e.g. big and small magnitudes), while still relying on learning absolute values[25]. Therefore, these absolute values cannot produce the seemingly irrational preferences observed in the transfer test.

The idea that the magnitude of available outcomes is somewhat rescaled by decision-makers is the cornerstone of the concept of utility[22]. In economics, this magnitude normalization is considered a stable property of individuals, and typically modeled with a marginally decreasing utility function whose parameters reflect individual core preferences[34,35] This approach was implemented in the UTILITY model, present in our model space. However, this model did not provide a satisfactory account of the behavioral data, and hence was not favored by the model-comparison approach. Similarly to the case of the POLICY model, this result derives from the fact that the UTILITY model cannot account for the emergence of reference-dependence, which is necessary to produce preference reversals between symbols of opposite valence in the transfer test. Crucially, correct choice rate during the learning sessions were equally well predicted by the UTILITY and the HYBRID models, thus highlighting the importance of using a transfer test, where options are extrapolated from original contexts, to challenge computational models of value learning and encoding[19,36,37].

Overall, our model comparison (based on both goodness-of-fit criteria and simulation-based falsification) favored the HYBRID model, which indicates that the pattern of choices exhibited by our subjects in the learning sessions and in the transfer test is most probably the result of a trade-off between absolute and relative values. In the HYBRID model, this trade-off was implemented by a subject-specific weight parameter ($\omega$), which quantified the relative influence of the normalized vs. absolute value-learning modules. A series of subsequent analyses revealed that several relevant factors affect this trade-off. First, we showed using an original trial-by-trial model comparison that the trade-off between absolute value-learning and normalized value learning implemented by the HYBRID model is progressive and gradual. This is an important novelty compared to previous work which only suggested such progressivity by showing that value rescaling was dependent of progressively acquired feedback information[19]. Note that learning normalized value ultimately converges to learning which option of a context is best, regardless of its valence or relative value compared to the alternative option. Second, and in line with the idea that information concerning the forgone outcome promotes state-dependent valuation[18,32], we also found that the relative weight of the normalized-value learning module ($\omega$) increased when more information was available (counterfactual feedback). Finally, individuals whose pattern of choices was indicative of a strong influence of the normalized value learning module (i.e., with higher $\omega$) appeared to have a better understanding of the task, assessed in the debriefing. Future research, using larger sample sizes and more diversified cohorts, will indicate whether or not the weight parameter (and therefore the value contextualization process) is useful to predict real life outcomes in terms of socio-economics achievements and psychiatric illness.

Overall, these findings suggest that value normalization is the results of a 'high-level'—or 'model-based'—process through which outcome information is not only used to update action values, but also to build an explicit representation of the embedding context where outcomes are experienced. Consistent with this interpretation, value normalization has recently been shown to be degraded by manipulations imposing a penalty for high-level costly cognitive functions, such as high memory load conditions in economic decision-making tasks[38]. One can also speculate that value contextualization should be impaired under high cognitive load[39] and when outcome information is made unconscious[40]. Future research using multi-tasking and visual masking could address these hypotheses[41]. An additional feature of the design suggests that this value normalization is an active process. In our paradigm the different choice contexts were

presented in an interleaved manner, meaning that a subject could not be presented with the same context more than a few times in a row. Therefore, contextual effects could not be ascribed to slow and passive habituation (or sensitization) processes.

Although the present results, together with converging evidence in economics and psychology, concordantly point that state-dependent valuation is needed to provide a satisfactory account of human behavior, there is still an open debate concerning the exact implementation of such contextual influences. In paradigms where subjects are systematically presented with full feedback information, it would seem that subjects simply encode the difference between obtained and forgone outcome, thus parsimoniously achieving full context-dependence without explicitly representing and encoding state value[18,32]. However, such models cannot be easily and effectively adapted to tasks where only partial feedback information is available. In these tasks, context-dependence has been more efficiently implemented by assuming separate representational structures for action and state values which are then used to center action-specific prediction errors[19,20]. In the present paper, we implemented this computational architecture in the HYBRID model, which builds on a partial adaptation scheme between an ABSOLUTE and a RELATIVE model. Although descriptive by nature, such hybrid models are commonly used in multi-step decision-making paradigms, e.g., to implement trade-offs between model-based and model free learning[42–44], because they allow to readily quantify the contributions of different learning strategies, and to straightforwardly map to popular dual-process accounts of decision-making[45,46]. In this respect, future studies adapting the present paradigm for functional imaging will be crucial to assess whether absolute and relative (i.e., reference-point centered and range adapted) outcome values are encoded in different regions (dual valuation), or whether contextual information is readily integrated with outcome values in a single brain region (partial adaptation). However, it should be noted that previous studies using similar paradigms, consistently provided support for the second hypothesis, by showing that contextual information is integrated in a brain valuation system encompassing both the ventral striatum and the ventral prefrontal cortex, which therefore represent 'partially adapted' values[19,20,29]. This is corroborated by similar observations from electrophysiological recordings of single neurons in monkeys[26,27,47,48].

As in our previous study[19,28], we also manipulated outcome valence in order to create 'gain' and 'loss' decision frames. While focusing on the results related to the manipulation of outcome magnitude, which represented the novelty of the present design, we nonetheless replicated previous findings indicating that subjects perform equally well in both decision frames and that this effect is parsimoniously explained assuming relative value encoding. This robust result contradicts both standard reinforcement principles and behavioral economic results. In the context of animal learning literature, while Thorndike's famous law of effect parsimoniously predicts reward maximization in a 'gain' decision frame, it fails to explain punishment minimization in the 'loss' frame. Mower elegantly formalized this issue[49] ('how can a shock that is not experienced, i.e., which is avoided, be said to provide […] a source of […] satisfaction?') and proposed the two-factor theory that can be seen as an antecedent of our relative value-learning model. In addition, the gain/loss behavioral symmetry is surprising with respects to behavioral economic theory because it contradicts the loss aversion principle[17]. In fact, if 'losses loom larger than gains', one would predict a higher correct response rate in the 'loss' compared to the 'gain' domain in our task. Yet, such deviations to standard behavioral economic theory are not infrequent when decisions are based on experience rather than description[50], an observation referred to as the "experience/

description gap"[51,52]. While studies of the "experience/description gap" typically focus on deviations regarding attitude risky and rare outcomes, our and other groups' results indicate that a less documented but nonetheless—robust instance of the experience/description gap is precisely the absence of loss aversion[3,53].

To conclude, state-dependent valuation, defined as the combination of reference-point dependence and range-adaptation, is a double-edged sword of value-based learning and decision-making. Reference-point dependence provides obvious beneficial behavioral consequences in punishment avoidance contexts and range-adaptation allows to perform optimally when decreasing outcome magnitudes. The combination of these two mechanisms (implemented in the HYBRID model) is therefore accompanied with satisfactory learning performance in all proposed contexts. However, these beneficial effects on learning performance are traded-off against possible suboptimal preferences and decisions, when options are extrapolated from their original context. Crucially, our results show that state-dependent valuation remains only partial. As a consequence, subjects under-performed in the learning sessions relative to full context-dependent strategies (RELATIVE model), as well as in the transfer test relative to absolute value strategies (ABSOLUTE model). These findings support the idea that bounded rationality may not only arise from intrinsic limitations of the brain computing capacity, but also from the fact that different situations require different valuation strategies to achieve optimal performance. Given the fact that humans and animals often interact with changing and probabilistic environments, apparent bounded rationality may simply be the result of the effort for being able to achieve a good level of performance in a variety of different contexts. These results shed new light on the computational constraints shaping everyday reinforcement learning abilities in humans, most-likely set by evolutionary forces to optimally behave in ecological settings featuring both changes and regularities[36].

## Methods

**Experimental subjects**. We tested 60 subjects (39 females; aged $22.3 \pm 3.3$ years). Subjects were recruited via Internet advertising in a local mailing-list dedicated to cognitive science-related activities. We experienced no technical problems, so we were able to include all 60 subjects. Experiment 1 included 20 subjects. The sample size was chosen based on previous studies. Experiment 2 included 40 subjects: we doubled the sample size because Experiment 2 involved a more complex design with an additional factor (see below). The research was carried out following the principles and guidelines for experiments including human participants provided in the declaration of Helsinki (1964, revised in 2013). The local Ethical Committee approved the study and subjects provided written informed consent prior to their inclusion. To sustain motivation throughout the experiment, subjects were given a bonus dependent on the actual money won in the experiment (average money won: $3.73 \pm 0.27$, against chance $T(59) = 13.9$, $P < 0.0001$).

**Behavioral protocol**. Subjects performed a probabilistic instrumental learning task adapted from previous imaging and patient studies[19]. Subjects were first provided with written instructions, which were reformulated orally if necessary. They were explained that the aim of the task was to maximize their payoff and that seeking monetary rewards and avoiding monetary losses were equally important. For each experiment, subjects performed two learning sessions. Cues were abstract stimuli taken from the Agathodaimon alphabet. Each session contained four novel pairs of cues. The pairs of cues were fixed, so that a given cue was always presented with the same other cue. Thus, within sessions, pairs of cues represented stable choice contexts. Within sessions, each pair of cues was presented 20 times for a total of 80 trials. The four cue pairs corresponded to the four contexts (reward/big magnitude, reward/small magnitude, loss/big magnitude, and loss/small magnitude). Within each pair, the two cues were associated to a zero and a non-zero outcome with reciprocal probabilities (0.75/0.25 and 0.25/0.75). On each trial, one pair was randomly presented on the left and the right side of a central fixation cross. Pairs or cues were presented in a pseudo-randomized and unpredictable manner to the subject (intermixed design). The side in which a given cue was presented was also pseudo-randomized, such that a given cue was presented an equal number of times in the left and the right of the central cue. Subjects were required to select between the two cues by pressing one of the corresponding two buttons, with their left or right thumb, to select the leftmost or the rightmost cue, respectively, within a

3000 ms time window. After the choice window, a red pointer appeared below the selected cue for 500 ms. At the end of the trial, the cues disappeared and the selected one was replaced by the outcome ("+1.0€", "+0.1€", "0.0€", "−0.1€" or "−1.0€") for 3000 ms. In Experiment 2, in the complete information contexts (50% of the trials), the outcome corresponding to the unchosen option (counterfactual) was displayed. A novel trial started after a fixation screen (1000 ms, jittered between 500 and 1500 ms). After the two learning sessions, subjects performed a transfer test. This transfer test involved only the eight cues (2*4 pairs) of the last session, which were presented in all possible binary combinations (28, not including pairs formed by the same cue) (see also ref. [18]). Each pair of cues was presented four times, leading to a total of 112 trials. Instructions for the transfer test were provided orally after the end of the last learning session. Subjects were explained that they would be presented with pairs of cues taken from the last session, and that all pairs would not have been necessarily displayed together before. On each trial, they had to indicate which of the cues was the one with the highest value by pressing on the buttons as in the learning task. Subjects were also explained that there was no money at stake, but encouraged to respond as they would have if it were the case. In order to prevent explicit memorizing strategies, subjects were not informed that they would have to perform a transfer test until the end of the second (last) learning sessions. Timing of the transfer test differed from that of the learning sessions in that the choice was self-paced and in the absence of outcome phase. During the transfer test, the outcome was not provided in order not to modify the option values learned during the learning sessions. Between the leaning sessions and the transfer test subjects were interviewed in order to probe the extent of their explicit knowledge of the task's structure. More precisely the structured interview assessed: (1) whether or not the subjects were aware about the cues being presented in fixed pairs (choice contexts); (2) how many choice contexts they believed were simultaneously present in a learning session. The experimenter recorded the responses, but provided no feedback about their correctness in order to not affect subjects' performance in the transfer test.

**Model-free analyses**. For the two experiments, we were interested in three different variables reflecting subjects' learning: (1) correct choice rate (i.e., choices directed toward highest expected reward or the lowest expected loss) during the learning task of the experiment. Statistical effects were assessed using multiple-way repeated measures ANOVAs with feedback valence, feedback magnitude, and feedback information (in Experiment 2 only) as within-subject factors; (2) correct choice rate during the transfer test, i.e., choosing the option with the highest absolute expected value (each symbol has a positive or negative absolute expected value, calculated as Probability(outcome) × Magnitude(outcome)); and (3) choice rate of the transfer test (i.e., the number of times an option is chosen, divided by the number of times the option is presented). The variable represents the value attributed to one option, i.e., the preference of the subjects for each of the symbols. Transfer test choice rates were submitted to multiple-way repeated measures ANOVAs, to assess the effects of option favorableness (being the most advantageous option of the pair), feedback valence and feedback magnitude as within-subject factors. In principle, probabilistic designs like ours the theoretical values (i.e., imposed by design) of the contexts and options may not correspond to the outcomes experienced by subjects. To verify that our design-based categories used in the ANOVAs analyses were legitimated, we checked the correlation between the theoretical and the empirical values of the outcomes. The results indicate that there was no systematic bias ($R > 0.99$; and $0.9 < $ slope $< 1.2$). Post-hoc tests were performed using one-sample $t$-tests. To assess overall performance, additional one-sample $t$-tests were performed against chance level (0.5). Correct choice rates from the learning test meet a normal distribution assumption (Kolmogorov–Smirnov test: $K(60) = 0.087$, $P > 0.72$; Lilliefors test: $K(60) = 0.087$, $P > 0.30$), as well as correct choice rates from the transfer test (Kolmogorov–Smirnov test: $K(60) = 0.092$, $P > 0.65$; Lilliefors test: $K(60) = 0.092$, $P > 0.22$). All statistical analyses were performed using Matlab (www.mathworks.com) and R (www.r-project.org).

**Model space**. We analyzed our data with extensions of the Q-learning algorithm[4,54]. The goal of all models was to find in each choice context (or state) the option that maximizes the expected reward $R$.

At trial $t$, option values of the current context $s$ are updated with the Rescorla–Wagner rule[5]:

$$\begin{aligned} Q_{t+1}(s, c) &= Q_t(s, c) + \alpha_c \delta_{c,t} \\ Q_{t+1}(s, u) &= Q_t(s, u) + \alpha_u \delta_{u,t} \end{aligned} \qquad (4)$$

where $\alpha_c$ is the learning rate for the chosen ($c$) option and $\alpha_u$ the learning rate for the unchosen ($u$) option, i.e., the counterfactual learning rate. $\delta_c$ and $\delta_u$ are prediction error terms calculated as follows:

$$\begin{aligned} \delta_{c,t} &= R_{c,t} - Q_t(s, c) \\ \delta_{u,t} &= R_{u,t} - Q_t(s, u) \end{aligned} \qquad (5)$$

$\delta_c$ is updated in both partial and complete feedback contexts and $\delta_u$ is updated in the complete feedback context only (Experiment 2, only).

We modeled subjects' choice behavior using a softmax decision rule representing the probability for a subject to choose one option $a$ over the other option $b$:

$$P_t(s, a) = \frac{1}{1 + e^{\left(\frac{Q_t(s,b) - Q_t(s,a)}{\beta}\right)}} \qquad (6)$$

where $\beta$ is the temperature parameter. High temperatures cause the action to be all (nearly) equi-probable. Low temperatures cause a greater difference in selection probability for actions that differ in their value estimates[4].

We compared four alternative computational models: the ABSOLUTE model, which encodes outcomes in an absolute scale independently of the choice context in which they are presented; the RELATIVE model which encodes outcomes on a binary (correct/incorrect) scale, relative to the choice context in which they are presented[55]; the HYBRID model, which encodes outcomes as a weighted sum of the absolute and relative value; the POLICY model, which encodes outcome in an absolute scale, but implements divisive normalization in the policy.

**ABSOLUTE model**. The outcomes are encoded as the subjects see them as feedback. A positive outcome is encoded as its "real" positive value (in euros) and a negative outcome is encoded as its "real" negative value (in euros): $R_{\text{ABS},t} \in \{-1.0\text{€}, -0.1\text{€}, 0.0\text{€}, 0.1\text{€}, 1.0\text{€}\}$.

**RELATIVE model**. The outcomes (both chosen and unchosen) are encoded on a context-dependent correct/incorrect relative scale. The model assumes the effective outcome value to be adapted to the range of the outcomes present in a given context. The option values are no longer calculated in an absolute scale, but relatively to their choice context value: in the delta-rule, the correct option is updated with a reward of 1 and the incorrect option is updated with a reward of 0. To determine the context of choice, the model uses a state value $V(s)$ stable over trials, initialized to 0, which takes the value of the first non-zero (chosen or unchosen) outcome in each context $s$.

$$R_{\text{REL},t} = \frac{R_{\text{ABS},t}}{|V_t(s)|} + \max\left\{0, \frac{-V_t(s)}{|V_t(s)|}\right\} \qquad (7)$$

Thus, the outcomes (chosen and unchosen) are now normalized to a context-dependent correct/incorrect encoding: $R_{\text{REL},t} \in \{0, 1\}$. The chosen and unchosen option values and prediction errors are updated with the same rules as in the ABSOLUTE model.

**HYBRID model**. At trial $t$ the prediction errors of the chosen and unchosen options are updated as a weighted sum of the absolute and relative outcomes:

$$R_{\text{HYB},t} = \omega * R_{\text{REL},t} + (1 - \omega) * R_{\text{ABS},t} \qquad (8)$$

where $\omega$ is the individual weight. At each trial $t$, the model independently encodes both outcomes as previously described and updates the final HYBRID outcome:

$$R_{\text{HYB},t} = \begin{cases} R_{\text{ABS},t} & \text{if } \omega = 0 \\ R_{\text{REL},t} & \text{if } \omega = 1 \end{cases}$$

The chosen and unchosen option values and prediction errors are updated with the same rules as in the ABSOLUTE model. If the RELATIVE model is conceptually similar to a policy-gradient algorithm, because it does not encode cardinal option values but only context-dependent ordinal preferences, the HYBRID model is reminiscent of a recently proposed model that features an interaction between a Q-learning and an actor-critic[56,57].

**UTILITY model**. We also considered a fourth UTILITY model, which implements the economic notion of marginally decreasing subjective utility at the outcome encoding step[17,22]. The big magnitude outcomes ($|R| = 1$) are re-scaled with a multiplicative factor $0.1 < v < 1.0$:

$$R_{\text{UTY},t} = v * R_{\text{ABS},t} \text{ if } |R| = 1 \qquad (9)$$

**POLICY model**. Finally, we considered a fith POLICY model that encodes option values as the ABSOLUTE model and normalizes them in the softmax rule, i.e., at the decision step only[25,26,47]:

$$P_t(s, a) = \frac{1}{1 + e^{\left(\frac{Q_t(s,b) - Q_t(s,a)}{Q_t(s,b) + Q_t(s,a)} * \frac{1}{\beta}\right)}} \qquad (10)$$

Additional computational hypotheses are addressed (and rejected) in the Supplementary Methods.

**Model fitting, comparison, and simulation**. Specifically for the learning sessions, transfer test, and both, we optimized model parameters, the temperature $\beta$, the factual learning rate $\alpha_F$, the counterfactual learning rate $\alpha_C$ (in Experiment 2 only) and the weight $\omega$ (in the HYBRID model only), by minimizing the negative log likelihood $LL_{\max}$ using Matlab's *fmincon* function, initialized at starting points of 1 for the temperature and 0.5 for the learning rates and the weight. As a quality check we replicated this analysis using multiple starting points and this did not change the results (Supplementary Table 2). We computed at the individual level the BIC using, for each model, its number of free parameters $d_f$ (note that the Experiment 2 has an additional parameter $\alpha_C$) and the number of trials ntrials (note that this number of trials varies with the optimization procedure: learning sessions only, 160, transfer test only, 112, or both, 272):

$$\text{BIC} = 2 * LL_{\max} + \log(\text{ntrials}) * d_f \qquad (11)$$

Model estimates of choice probability were generated trial-by-trial using the optimal individual parameters. We made comparisons between predicted and actual choices with a one-sample $t$-test and tested models' performances out of the sample by assessing their ability to account for the transfer test choices. On the basis of model-estimate choice probability, we calculated the log-likelihood of learning sessions and transfer test choices that we compared between computational models. Finally, we submitted the model-estimate transfer-test choice probability to the same statistical analyses as the actual choices (ANOVA and post-hoc $t$-test; within-simulated data comparison) and we compared modeled choices to the actual data. In particular, we analyzed actual and simulated correct choice rates (i.e., the proportions of choices directed toward the most advantageous stimulus) and compared transfer-test choices for each symbol with a sampled $t$-test between the behavioral choices and the simulated choices.

**Code availability**. All custom scripts have been made available from Github repository https://github.com/sophiebavard/Magnitude. Additional modified scripts can be accessed upon request.

## Data availability
Data that support the findings of this study are available from Github repository https://github.com/sophiebavard/Magnitude.

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

## Acknowledgements

Emmanuel Noblins and Alexander Salvador provided help for data collection. S.P. is supported by an ATIP-Avenir grant (R16069JS) Collaborative Research in Computational Neuroscience ANR-NSF grant (ANR-16-NEUC-0004), the Programme Emergence (s) de la Ville de Paris, and the Fondation Fyssen. S.B. is supported by MILDECA (*Mission Interministérielle de Lutte contre les Drogues et les Conduites Addictives*) and the EHESS (*Ecole des Hautes Etudes en Sciences Sociales*). M.L. is supported by an NWO Veni Fellowship (Grant 451-15-015) and a Swiss National Fund Ambizione grant (PZ00P3_174127). The Institut d'Etudes de la Cognition is supported financially by the LabEx IEC (ANR-10-LABX-0087 IEC) and the IDEX PSL* (ANR-10-IDEX-0001-02 PSL*). The funding agencies did not influence the content of the manuscript.

## Author contributions

S.P. and G.C. designed the task. S.P. performed the experiments. S.B., M.L., and S.P. analyzed the data. S.B., M.L., S.P., and M.K. wrote the manuscript. All authors interpreted the results, commented, and approved the final version of the manuscript.

## Additional information

**Competing interests:** The authors declare no competing interests.

