## [Peer Review File · Nature Communications]

Reviewers' comments:

Reviewer #1 (Remarks to the Author):

In this article, the authors investigate how adult human participants integrate feedback at various scales and valences when learning through reinforcement. They show behavioral and modeling evidence that participants take into account the reward context to scale the received reward before integrating it into value computations in reinforcement learning algorithms.

This is a well written paper and with clean modeling and rigorous analyses. It covers an interesting question. I have two suggestions that should improve the paper.

1. Other articles have explored similar questions but are not referenced – this is unfortunate because they propose an alternative, competing mechanism by which the interesting behavioral results could occur. For example, Gold et al (2012, Arch Gen Psych) suggest that this kind of phenomenon can be modeled by mixing an actor critic module with an RL. Indeed, the actor critic captures the context-dependent policy (item A is better than item B in pair AB), but not the relative values of the items across pairs; while the Q-learner does. This might provide a competing mechanism for capturing the main effects described l143-145. Comparing the two accounts is important.

2. Analyzing the test phase results as a function of the experimenter-defined conditions can lead to biased results. Indeed, "incorrect" options during the learning phase are selected less often, and thus experience fewer feedback signals than correct options. As such, the learned values are noisier and might not match the planned experimental values; this can negatively influence the analysis. A better way to proceed is to use the actual empirical experienced outcomes as a dependent variable, and use logistic regression with this continuous variable, instead of anovas on predefined conditions. See e.g. Schutte et al (2017, Plos One), for an example of the methods, and the potential confounds of not using this method.

Minor comments

L143/145 is hard to parse, but is the crucial result. It would be helpful to plot the results specifically testing these crucial results, with a schematic of what the test is.

Reviewer #2 (Remarks to the Author):

The researchers apply computational modeling to a probabilistic choice task to examine how contextual factors effect valuation during reinforcement learning. This extends the group's previous work (Palminteri et al., 2014, Nature Comms) in which the authors demonstrated that the state context determines the reference point for evaluating outcomes (for instance, "reference-point dependence" helps to explain how avoiding a loss in the context of frequent losses might be experienced as positive). Here, the researchers probe reference-point dependence as well as range adaptation (i.e., how evaluation of outcomes scales based on the relative magnitude of available options) by independently manipulating outcome valence (gains, losses) and magnitude (big, small).

They find that the computational model that best fits subjects' behavior invokes a hybrid of a standard Q-learning reinforcement learning that learns the absolute values of options coupled with a reinforcement learning model that learns only the relative values of presented options (or which of two options is better, regardless of outcome valence or magnitude). They additionally demonstrate that performance initially conforms more to predictions of the absolute model but then later shifts to conform more to predictions of the relative model. Finally, they quantitatively compare the

performance of the hybrid model to other models implied by standard psychology and economics literatures in which normalization does not gradually emerge over time, but is instantiated at the time of choice (policy and utility models) and find the hybrid model to be superior. This is high quality behavioral work which comprehensively provides model-free evidence and model-based comparisons to provide key evidence for the superiority of a hybrid model of valuation that should interest a broad range of scientists, including psychologists, neurologists, psychiatrists, economists, laypeople and others.

The work has many positive features, including establishing a ground truth with model free assessment, and then quantitative comparisons of several well-justified models, deep analysis of how model fits change over time, and consideration of the usefulness (or rationality) of these strategies in different learning regimes. The findings add a novel feature of testing different accounts of range adaptation (currently a hot topic) in addition to relative valuation. Novel implications include the finding that the reliance on the absolute versus relative strategies shifts over time, and observation that reliance on one strategy (e.g., relative valuation) can serve the subject well in some scenarios (e.g., learning) but poorly in others (e.g., transfer), creating apparent "inconsistencies" in choice. The findings hold obvious implications for how to model valuation and future studies trying to deconstruct the neural bases of these processes (as noted by the authors).

The paper is so thorough and convincing that I have few suggestions for improvement. The findings are presented as if the observed patterns apply to most or all subjects, and I am willing to believe this is true, but I found myself wondering about the heterogeneity of individuals with respect to the absolute versus relative valuation strategy. I personally find the Methods last organization more confusing, but leave that determination to the authority of the editors. Specific suggestions are listed below in order of appearance:

Line 291: Should "outcome valence" read "outcome magnitude"?

Line 550: The authors might also note that this rule is applied at the decision step (if true).

Reviewer #3 (Remarks to the Author):

This is a straightforward paper that is an extension of earlier work trying to carefully pin down deviations from "absolute" (i.e state- or context-independent) valuation in simple learning and choice. There are a lot of good clear results here. I particularly like the link between REL weight omega and explicit task knowledge.

One major comment is that in the REL scaled valuation, stimuli outcomes are scaled to 0-1 according to "win/loss" within a pair. This combines reference-dependence (the best outcome is compared to the alternative a la disappointment) and range-adaptation. Can't these be separated in some way? I realize the transfer test separates the different effects in later choices behaviorally, but it would also be nice to be able to model them separately.

One thing good to see is noted in * below (comparisons in transfer between all A-H pairs).

Figure 1 is nice and clear.

Line 556 Experience is a typo

Line 158 and others. I don't understand the notation. Shouldn't each stimulus A to H have a separate R value that changes over time? There is no variable or subscript indexing stimulus letter. Also what does it mean that V(s) is "initialized"? Does it change over time?

*Line 143. This indicates that (.1,75%) C is chosen 71% in all pairs than B=(1,25%) right? These choice frequencies are likely to vary when C and B are compared with the other stimuli in pairwise choice. This is additional information you should report (E.g. a table reporting all pairwise choice frequencies. Particularly, it would be good to know what happens when the pair B and C are compared in the transfer test.

Line 291. Should be outcome magnitude not valence, right?

Line 330 paragraph. I think this is quite wrong. In modern consumer theory indifference curves are independent of choice sets; utilities are (ordinally) ranked so that no context effects are present. Put differently, in making choices from disparate items a lagrangian multiplier represents the utility value of a marginal dollar. If you add in say cheap or expensive goods that will not be purchased their utility values do not exhibit any reference or normalization effect.

Line 418 is quite a bold claim. I agree with the spirit of the last paragraph, that adaptation is useful for learning and bad for later choice (based on learned values). But the idea that foraging is the only adapted problem is quite a stretch: Especially in primates and humans, there is a wide range of learning and state adaptations that might be needed (climate, seasonality in food and fertility, etc.)

P 548. How is the utility nonlinearity used? Is it an input to other models? If I understand it, adding this feature to the HYBRID model acts as partial range-adaptation.

We thank the Reviewers for their interest in our work and positive comments on the manuscript. The Reviewers made relevant suggestions and we appreciate the opportunity to improve the manuscript accordingly. Please find below point-by-point responses.

Reviewer #1 (Remarks to the Author):

In this article, the authors investigate how adult human participants integrate feedback at various scales and valences when learning through reinforcement. They show behavioral and modeling evidence that participants take into account the reward context to scale the received reward before integrating it into value computations in reinforcement learning algorithms. This is a well-written paper and with clean modeling and rigorous analyses. It covers an interesting question. I have two suggestions that should improve the paper.

R1.1

Other articles have explored similar questions but are not referenced – this is unfortunate because they propose an alternative, competing mechanism by which the interesting behavioral results could occur. For example, Gold et al (2012, Arch Gen Psych) suggest that this kind of phenomenon can be modeled by mixing an actor critic module with an RL. Indeed, the actor critic captures the context-dependent policy (item A is better than item B in pair AB), but not the relative values of the items across pairs; while the Q-learner does. This might provide a competing mechanism for capturing the main effects described l143-145. Comparing the two accounts is important.

We thank Reviewer 1 for pointing out the omitted reference, that we now included in the revised manuscript (reference number 56). We agree that our model is conceptually similar to the actor-critic (AC) architecture. We actually consider the learning processes and the principles behind the AC architecture not as a competing mechanism, but rather as alternate way to implement the notion of context dependence-reinforcement learning. We have now clarified our views in the methods (page 17):

If the RELATIVE model is conceptually similar to a policy-gradient algorithm, because it does not encode cardinal option values but only context-dependent ordinal preferences, the HYBRID model is reminiscent of a recently proposed model that features an interaction between a Q-learning and an actor-critic^{56,57}.

Figure R1 (Figure S1): generative performance of the HYBRID model compared to the Q-learning/Actor-Critic hybrid proposed by Gold et al.

Additionally, following the Reviewer 1's explicit recommendation, we explicitly compared the model proposed in Gold et al (2012) to our HYBRID model. Relative model comparison favoured the Hybrid model ($T(59)=4.80$, $P<0.0001$; see **Supplementary Table 1**).

Model simulation analysis (**Figure R1**) showed that while Gold et al's model matched the effect in the learning test, it comparably failed to capture the value inversions in the transfer test. A closer look suggests that it specifically missed value inversion across positive and negative valence. We now present these results in the revised supplementary materials (page 1 & **Figure S1**).

R1.2

Analyzing the test phase results as a function of the experimenter-defined conditions can lead to biased results. Indeed, "incorrect" options during the learning phase are selected less often, and thus experience fewer feedback signals than correct options. As such, the learned values are noisier and might not match the planned experimental values; this can negatively influence the analysis. A better way to proceed is to use the actual empirical experienced outcomes as a dependent variable, and use logistic regression with this continuous variable, instead of anovas on predefined conditions. See e.g. Schutte et al (2017, Plos One), for an example of the methods, and the potential confounds of not using this method.

We agree with the Reviewer 1 that trial-by-trial dynamics are often relevant for model-free in reinforcement learning data. However, the goal of the model-free analysis in the present study was to assess whether or not contextual factors (especially outcome magnitude, as imposed by the design) affect overall performance in the learning test, as measured by average correct response rate, and overall option preference, as measured by average rate in the transfer test. The choice of these metrics was not arbitrary but justified *by practice* (we used the very same analytical pipeline in Palminteri et al. 2015) and *by principle*, because the models of interest predict radically different behaviours when projected on to these metrics.

However, Reviewer 1 is correct arguing that in probabilistic designs like ours the planned experimental values of the contexts and options may not correspond to the empirical experienced outcomes. To address this issue we calculated the correlation between the *planned* experimental values of the contexts/options and the average empirical experienced outcomes. Luckily, the results (reported below in **Figure R2**) indicate that there was no systematic bias: the correlations are very strong ($R>0.99$) and indistinguishable from the identity ($0.9<\text{slope}<1.2$). This is due to the fact that our design did not involve very rare events (the rarer outcome was still presented 25% of the times) and that subjects sufficiently sampled both options (average correct response rate ~70%). In the methods section of the revised manuscript we now mention this important control (page 15):

"In principle, probabilistic designs like ours the theoretical values (i.e., imposed by design) of the contexts and options may not correspond to the outcomes experienced by subjects. To verify that our design-based categories used in the ANOVAs analyses were legitimated, we checked the correlation between the theoretical and the empirical values of the outcomes. The results indicate that there was no systematic bias ($R>0.99$; and $0.9<\text{slope}<1.2$)."

Finally, in order to further justify the utilisation of an ANOVA approach, we verified the assumption of normality in the dependent variables of interest (which is not guaranteed since

choice rates are bounded between 0 and 1), with Kolmogorov-Smirnov and Lilliefors tests. Results were consistent with normal distributions in both the learning ($K(60)=0.087$, $P>0.72$; Lilliefors test: $K(60)=0.087$, $P>0.30$) and the transfer test ($K(60)=0.092$, $P>0.65$; Lilliefors test: $K(60)=0.092$, $P>0.22$). We added this information in methods section of the revised manuscript (page 15)

“Correct choice rates from the learning test meet a normal distribution assumption (Kolmogorov-Smirnov test: $K(60)=0.087$, $P>0.72$; Lilliefors test: $K(60)=0.087$, $P>0.30$), as well as correct choice rates from the transfer test (Kolmogorov-Smirnov test: $K(60)=0.092$, $P>0.65$; Lilliefors test: $K(60)=0.092$, $P>0.22$).”

Figure R2: correlation between the *planned* and the *empirical* contexts and options values indicate no systematic bias. Note that error bars represent standard deviation of the mean.

R1.3

Minor comments

L143/145 is hard to parse, but is the crucial result. It would be helpful to plot the results specifically testing these crucial results, with a schematic of what the test is.

Thank for this suggestion (shared with Reviewer 3). In the revised manuscript we added the results of the pairwise (i.e., cue-by-cue) choice rate (page 5):

To verify that these value inversions were not only observed at the aggregate level (i.e., were not an averaging artifact), we analyzed the transfer test choice rate for each possible comparison. Crucially, analysis of the pairwise choices confirm value inversion also for direct comparisons (Figure 3A).

As Reviewer 1 correctly points out, this is the crucial results, we therefore present it in a new main figure (Figure 3, see bellows), where we also show that the behavioural patter is fully compatible with the HYBRID model.

Figure R3 (and Figure 3): Transfer test behavioral results and model simulations. Colored map of pairwise choice rates during the transfer test for each symbol when compared to each of the seven other symbols. Comparisons between the same symbols are undefined (black squares). (A) Experimental data. (B) ABSOLUTE model. (C) HYBRID model. (D) RELATIVE model.

Reviewer #2 (Remarks to the Author) :

The researchers apply computational modeling to a probabilistic choice task to examine how contextual factors effect valuation during reinforcement learning. This extends the group's previous work (Palminteri et al., 2014, Nature Comms) in which the authors demonstrated that the state context determines the reference point for evaluating outcomes (for instance, "reference-point dependence" helps to explain how avoiding a loss in the context of frequent losses might be experienced as positive). Here, the researchers probe reference-point dependence as well as range adaptation (i.e., how evaluation of outcomes scales based on the relative magnitude of available options) by independently manipulating outcome valence (gains, losses) and magnitude (big, small).

They find that the computational model that best fits subjects' behavior invokes a hybrid of a standard Q-learning reinforcement learning that learns the absolute values of options coupled with a reinforcement learning model that learns only the relative values of presented options (or which of two options is better, regardless of outcome valence or magnitude). They additionally demonstrate that performance initially conforms more to predictions of the absolute model but then later shifts to conform more to predictions of the relative model. Finally, they quantitatively compare the performance of the hybrid model to other models implied by standard psychology and economics literatures in which normalization does not gradually emerge over time, but is instantiated at the time of choice (policy and utility models) and find the hybrid model to be superior. This is high quality behavioral work which comprehensively provides model-free evidence and model-based comparisons to provide key evidence for the superiority of a hybrid model of valuation that should interest a broad range of scientists, including psychologists, neurologists, psychiatrists, economists, laypeople and others.

The work has many positive features, including establishing a ground truth with model free assessment, and then quantitative comparisons of several well-justified models, deep analysis of how model fits change over time, and consideration of the usefulness (or rationality) of these strategies in different learning regimes. The findings add a novel feature of testing different accounts of range adaptation (currently a hot topic) in addition to relative valuation. Novel implications include the finding that the reliance on the absolute versus relative strategies shifts over time, and observation that reliance on one strategy (e.g., relative valuation) can serve the subject well in some scenarios (e.g., learning) but poorly in others (e.g., transfer), creating apparent "inconsistencies" in choice. The findings hold obvious implications for how to model valuation and future studies trying to deconstruct the neural bases of these processes (as noted by the authors).

R2.1

The paper is so thorough and convincing that I have few suggestions for improvement. The findings are presented as if the observed patterns apply to most or all subjects, and I am willing to believe this is true, but I found myself wondering about the heterogeneity of individuals with respect to the absolute versus relative valuation strategy.

As the reviewer correctly points out the effect is present in almost the individual. If we take the hybridization parameter (ω) as a summary measure of our effect of interest we do not detect any sign of bi- or multi-modality (see **Figure R4**, below). For illustrative purposes we show the results of the transfer test in subjects with $\omega < 0.5$ and $\omega > 0.5$. Crucially, even in these rather 'extreme' groups we found that the value contextualization process (range-

adaptation and reference point dependence) is partial: subjects with $\omega < 0.5$ still present value inversions and subjects with $\omega > 0.5$ still present a valence effect (see **Figure R4**).

However, it would be unfair to say that there is no sign of inter-individual variability in this computational process. Importantly, as showed in the Figure below, this variability is quite well captured by the hybrid model (and the ω parameter). Indeed, in the manuscript we show that variability in the ω parameter presents some extent of external validity, being predictive of declarative understanding of the task structure. Taken together these results suggest that our paradigm show promises to investigate the computational bases of inter-individual differences in terms of real life outcomes (a project that is part of our future agenda). We mention this perspective in the discussion of the revised manuscript (page 11):

“Future research, using larger sample sizes and more diversified cohorts, will indicate whether or not the weight parameter (and therefore the value contextualization process) is useful to predict real life outcomes in terms of socio-economics achievements and psychiatric illness.”

Figure R4: the black/white histograms represent the weight parameter for each subject. The inset represents the transfer test choice rate per cue (data: bars; grey points: model-simulations) in subject with low (left) and high (right) weight parameter.

R2.2

I personally find the Methods last organization more confusing, but leave that determination to the authority of the editors. Specific suggestions are listed below in order of appearance:

We followed a very similar approach to that of previous published studies in journals where the methods appear after the discussion by providing the essential of the task and the model space in the result session. Concerning the methods sessions per se, we followed a standard organization for behavioural-computational studies (Palminteri et al. 2016; Palminteri et al. 2017). Following this first round of revisions several aspects have been improved, concerning the modelling and the statistical analyses part. We hope the overall clarity of the methods has improved in a satisfactory manner.

R2.3

Line 291: Should “outcome valence” read “outcome magnitude”?

Thank you for spotting this, we have corrected accordingly.

R2.4

Line 550: The authors might also note that this rule is applied at the decision step (if true).

In the model space only the POLICY model normalizes at the decision step. In the UTILITY model decreasing marginal utility is applied at the reward prediction error step (necessary to explain transfer test pattern). We clarified in the methods for each model at which step (decision or learning) value normalization applies.

Reviewer #3 (Remarks to the Author):

This is a straightforward paper that is an extension of earlier work trying to carefully pin down deviations from “absolute” (i.e state- or context-independent) valuation in simple learning and choice. There are a lot of good clear results here. I particularly like the link between REL weight ω and explicit task knowledge.

R3.1

One major comment is that in the REL scaled valuation, stimuli outcomes are scaled to 0-1 according to “win/loss” within a pair. This combines reference-dependence (the best outcome is compared to the alternative a la disappointment) and range-adaptation. Can’t these be separated in some way? I realize the transfer test separates the different effects in later choices behaviorally, but it would also be nice to be able to model them separately.

We agree that there is no a priori reason to suppose that range adaptation and reference point dependence are governed by the same parameter and in principle they could be dissociable phenomena. To assess this hypothesis we considered an alternative model (the SEPARATE model), where the two processes are governed by different parameters (ρ for range adaptation and π for reference point dependence). The model is designed so that if both parameters are set to 0 (or to 1), the model is equivalent to the ABSOLUTE (or the RELATIVE) model. Unsurprisingly, inspection of model simulations of the SEPARATE model indicates that the model was perfectly capable to reproduce all features of subjects’ behaviour, since the RELATIVE model is nested within the SEPARATE model (**Figure R5 – S1**). However, when we assessed the parsimony of the RELATIVE model compared to the HYBRID model, we found that relative model comparison favoured the HYBRID model ($T(59)=5.42$, $P<0.0001$). In addition, consistent with the idea that range adaptation and reference point dependence are – at least in part – related phenomena, we retrieved a significant correlation between the ρ and the π parameter ($R= 0.31$, $P<0.02$). Of course from our data and results we are claiming that the two phenomena are manifestation of the same process as the RELATIVE model implies, but rather than that we have no evidence for the opposite hypotheses. In this respect future studies involving imaging data will prove particularly useful.

We included the results involving the SEPARATE model in the supplementary materials of the revised manuscript (page 1).

Figure R5 (Figure S1): generative performance of the HYBRID model compared to the SEPARATE model proposed by the Reviewer 3.

R3.2

Line 556 Experience is a typo

Thank you for spotting this.

R3.3

Line 158 and others. I don't understand the notation. Shouldn't each stimulus A to H have a separate R value that changes over time? There is no variable or subscript indexing stimulus letter. Also what does it mean that $V(s)$ is "initialized"? Does it change over time?

Thank you for giving the opportunity of clarifying this point. R_{ABS} is the experienced outcome in an absolute scale (i.e., -1.0€, -0.1€, 0.0€, 0.1€, and 1.0€, depending on the context and stimulus). Unbeknown to the participants, R_{ABS} values are specifically bounded to specific learning contexts. In the RELATIVE model, Q-values are learned using R_{REL} , which is R_{ABS} with range-adaptation and reference-point dependence (in a binary scale).

It is true that we should have put t in $V(s)$ in equation (1), thank for spotting this (corrected in pages 5 & 6). As a matter of fact $V(s)$ are all initialized at 0.0 at the beginning of learning. Then, in the way we implemented context state value learning, it takes the value of the first non-zero observed outcome (R_{ABS}). We did not allow much more flexibility to the algorithm because, by design, in each context only one outcome type can be encountered. However, we do believe that in more dynamic environment, a more flexible version of context value learning should be implemented.

R3.4

*Line 143. This indicates that (.1,75%) C is chosen 71% in all pairs than B=(1,25%) right? These choice frequencies are likely to vary when C and B are compared with the other stimuli in pairwise choice. This is additional information you should report (E.g. a table reporting all pairwise choice

frequencies. Particularly, it would be good to know what happens when the pair B and C are compared in the transfer test.

Thank for this suggestion (shared with Reviewer 1). In the revised manuscript we added the results of the pairwise (i.e., cue-by-cue) choice rate (page 5):

To verify that these value inversions were not only observed at the aggregate level (i.e., were not an averaging artifact), we analyzed the transfer test choice rate for each possible comparison. Crucially, analysis of the pairwise choices confirm value inversion also for direct comparisons (Figure 3A).

As Reviewer 1 correctly points out, this is the crucial results, we therefore present it in a new main figure (Figure 3, see bellows), where we also show that the behavioural patter is fully compatible with the HYBRID model.

Figure R6 (and Figure 3): Transfer test behavioral results and model simulations. Colored map of pairwise choice rates during the transfer test for each symbol when compared to each of the seven other symbols. Comparisons between the same symbols are undefined (black squares). (A) Experimental data. (B) ABSOLUTE model. (C) HYBRID model. (D) RELATIVE model.

R3.5

Line 291. Should be outcome magnitude not valence, right?

Thank you for spotting this, we corrected the typo.

R3.6

Line 330 paragraph. I think this is quite wrong. In modern consumer theory indifference curves are independent of choice sets; utilities are (ordinally) ranked so that no context effects are present. Put differently, in making choices from disparate items a lagrangian multiplier represents the utility value of a marginal dollar. If you add in say cheap or expensive goods that will not be purchased their utility values do not exhibit any reference or normalization effect.

We agree with the reviewer that marginal decreasing utility function (as implemented) in the UTILITY model cannot account for contextual effect. However, we wanted to include this model as a control model because it is widely used in economics and we suspected that it was capable to reproduce – at least in part our crucial behavioural effects (partial magnitude effect in the learning test and value inversion in the transfer test). As a matter of fact it does so, but model simulation and BIC analyses indicate that it is outperformed by the HYBRID model. (see figure S1 below).

Figure R7 (Figure S1): generative performance of the HYBRID model compared to the UTILITY.

R3.7

Line 418 is quite a bold claim. I agree with the spirit of the last paragraph, that adaptation is useful for learning and bad for later choice (based on learned values). But the idea that foraging is the only adapted problem is quite a stretch: Especially in primates and humans, there is a wide range of learning and state adaptations that might be needed (climate, seasonality in food and fertility, etc.)

We are sorry for the misunderstanding; by referring to foraging behaviour we meant more general 'ecological settings'. We modified the last sentence accordingly (page 13):

These results shed new light on the computational constraints shaping everyday reinforcement learning abilities in humans, most-likely set by evolutionary forces to optimally behave in ecological settings featuring both changes and regularities³⁷.

R3.8

P 548. How is the utility nonlinearity used? Is it an input to other models? If I understand it, adding this feature to the HYBRID model acts as partial range-adaptation.

No, the utility nonlinearity is not an input to other models, we tested it in a separate specific model. Yes, similarly to the HYBRID model, the UTILITY model is able to capture the

performance difference between magnitude contexts in the learning test but fails to predict value inversion in the transfer test, thus the model is not parsimonious.

REVIEWERS' COMMENTS:

Reviewer #1 (Remarks to the Author):

I am satisfied with the authors' revision.

Reviewer #2 (Remarks to the Author):

The authors have comprehensively and constructively responded to the reviewers' feedback and deserve congratulations for this thorough and helpful contribution to the literature.

Reviewer #3 (Remarks to the Author):

R3 This response was good and I am glad you did some additional analyses to address points addressed by all referees.